# Trunk rotation, spinal deformity and appearance, health-related quality of life, and treatment adherence: Secondary outcomes in a randomized controlled trial on conservative treatment for adolescent idiopathic scoliosis

Marlene Dufvenberg[1]*, Anastasios Charalampidis[2,3], Elias Diarbakerli[2,3], Birgitta Öberg[1], Hans Tropp[4,5,6], Anna Aspberg Ahl[7], Hans Möller[2,8], Paul Gerdhem[2,9,10], Allan Abbott[1,6], on behalf of The CONTRAIS Study Group[¶]

1 Department of Health, Medicine and Caring Sciences, Unit of Physiotherapy, Linköping University, Linköping, Sweden, 2 Department of Clinical Science, Intervention and Technology (CLINTEC), Division of Orthopaedics and Biotechnology, Karolinska Institutet, Stockholm, Sweden, 3 Department of Reconstructive Orthopaedics, Karolinska University Hospital Huddinge, Stockholm, Sweden, 4 Department of Biomedical and Clinical Sciences, Linköping University, Linköping, Sweden, 5 Center for Medical Image Science and Visualization, Linköping University, Linköping, Sweden, 6 Department of Orthopaedics, Linköping University Hospital, Linköping, Sweden, 7 Department of Orthopaedics, Ryhov County Hospital, Jönköping, Sweden, 8 Stockholm Center for Spine Surgery, Stockholm, Sweden, 9 Department of Orthopaedics and Hand surgery, Uppsala University Hospital, Uppsala, Sweden, 10 Department of Surgical Sciences, Uppsala University, Uppsala, Sweden

¶ Membership of the CONTRAIS Study Group is provided in Supporting information.
* marlene.dufvenberg@liu.se

## Abstract

### Objective

To explore secondary outcomes at endpoint comparing treatments with adequate self-mediated physical activity combined with either night-time brace (NB), scoliosis-specific exercise (SSE), or adequate self-mediated physical activity alone (PA) in Adolescent Idiopathic Scoliosis (AIS).

### Methods

A longitudinal, prospective, multicenter RCT was conducted including 135 girls/boys, Cobb angle 25–40°, 9–17 years, and ≥1-year remaining growth were randomly allocated into NB, SSE, or PA group. Endpoint was curve progression of ≤6° (success) at skeletal maturity or >6° (failure). Outcomes included angle of trunk rotation (ATR), major curve Cobb angle, Spinal Appearance Questionnaire (pSAQ), Scoliosis Research Society-22r (SRS-22r), EQ-5Dimensions Youth 3Levels (EQ-5D-Y-3L), and EQ-Visual-Analogue-Scale (EQ-VAS), adherence to treatment and International Physical Activity Questionnaire (IPAQ-SF).

### Results

At endpoint, 122 patients were analyzed per protocol, mean age 12.7 (±1.4) years, and mean Cobb angle 31° (±4.3). A significant difference in change for ATR favored NB group

**Data availability statement:** Interested parties may contact ethics@lshtm.ac.uk to request access to the data. This approach has been adopted due to the sensitive nature of the data and the study population, as requested by the USJ Ethics Committee. Each request will be carefully assessed on a case-by-case basis to ensure ethical considerations and participant confidentiality are maintained. Further information is available from the Ethics Committee at the London School of Hygiene and Tropical Medicine (ethics@lshtm.ac.uk).

**Funding:** Paul Gerdhem; Grants: Region Stockholm (clinical research appointment). Allan Abbott & Paul Gerdhem; Grants: The Swedish Research Council (Dnr 521-2012-1771); the regional agreement on medical training and clinical research (ALF) between Stockholm County Council, Karolinska Institutet, and Linköpings University; and the Swedish Society of Spinal Surgeons. The funders had no role in study design, data collection and analysis, decision to publish, or preparation of the manuscript.

**Competing interests:** The authors have declared that no competing interests exist.

compared to SSE group -2.0º (95% CI -3.7 to -0.3). EQ-5D-Y-3L dimensions showed a significant difference in change with decrease in mobility (p=0.031), and usual activities (p=0.003) for SSE compared to NB and PA groups. Treatment adherence was adequate but slightly better in NB and PA groups compared to SSE on self-report (p=0.012), and health care provider (HCP) report was better in PA compared to SSE group (p=0.013). Higher motivation and capability explained 53% of the variability and gave better odds for higher adherence (OR = 11.12, 95% CI = 1.5 to 34.4; OR = 7.23, 95% CI = 2.9 to 43.3), respectively.

## Conclusions

Night-time brace, scoliosis-specific exercise or physical activity interventions for adolescents with idiopathic scoliosis showed small differences between groups in trunk rotation, spinal deformity and appearance, health-related quality of life, and treatment adherence but not likely reaching clinical relevance.

## Introduction

Conservative treatments with a combination of braces and exercises for adolescents with idiopathic scoliosis (AIS) aim to prevent or limit curve progression. Night-time bracing has shown comparable outcomes to full-time thoracolumbosacral orthoses, mainly for curves not exceeding 35° [1,2], but strength of conclusions is limited due to sample size and overall quality of included studies [3–5]. The potential advantages with night-time bracing are higher adherence, mitigated negative health effects, and less restrictions during daytime activities [2,6–9]. In the CONTRAIS study Charalampidis et al. [10] found that treatment with night-time bracing had higher success rate compared to control group with self-mediated physical activity alone.

Scoliosis-specific exercise [11] has shown promising results mainly in mild scoliosis in terms of perceived improved spine curvature, but limited evidence exists in reducing the Cobb angle [12–14]. A recent review concluded that rigorous studies are needed to evaluate the effectiveness of different exercise programs due to low quality of evidence [15]. A systematic review concluded [16] that there is a very low quality of evidence suggesting that bracing provides 2.7° better Cobb angle compared to scoliosis-specific exercise. This difference can however be considered within the standard error of measurement [17,18]. The CONTRAIS study Charalampidis et al. 10] concluded that scoliosis-specific exercise did not show higher success rate compared to control group with self-mediated physical activity alone in the prevention of curve progression in moderate AIS.

General physical activity is associated with numerous health benefits [19,20]. A minimum of 60 min of physical activity every day is recommended [21,22]. However, the majority (80%) of adolescents aged 11–17 years do not meet current physical activity guidelines [23]. Literature is not consistent regarding report on reduced physical ability levels in patients with AIS and association to scoliosis progression [24–28].

The current paper aims to explore endpoint secondary outcomes in the CONTRAIS trial of adequate self-mediated physical activity combined with either night-time brace (NB), scoliosis-specific exercise (SSE), or control with adequate self-mediated physical activity alone (PA) in moderate-grade AIS. Secondary outcomes included angle of trunk rotation, spinal deformity and appearance, health-related quality of life, and treatment adherence.

## Methods

### Study design and participants

This was a longitudinal, prospective, multicenter RCT. The protocol with detailed information is published on clinicaltrials.gov (NCT01761305) on 4/01/2013[29]. Approval was obtained by the Regional Ethical Board in Stockholm (Dnr 2012/172–31/4, 2015/1007–32, and 2017/609–32).

Girls and boys 9–17 years of age with idiopathic scoliosis 25–40° Cobb angle [30, 31] and estimated remaining growth of at least one year were included [32]. Exclusion criteria were previous treatment for scoliosis, or inability to understand Swedish. Consecutive recruitment took place from start date 08-01-2013 to end date 23-10-2018 at 6 study centers (Karolinska University Hospital, Linköping University Hospital, Ryhov Hospital Jönköping, Eskilstuna Hospital, Västerås Hospital, Sundsvall Hospital). Eligibility for participation was decided at each study center where radiographs were further assessed at the principal investigation center by two experienced spine surgeons. Patients that declined participation were offered standard care with a corrective thoracolumbosacral Boston brace, worn at least 20 hours per day.

### Interventions

A summary of the three treatment groups is provided below with detailed information in Supporting information, and previously published protocol [29]. Similarity for all allocated treatments were to enhance high adherence to treatment plan and self-management. Methods used can be described in terms of Capability, Opportunity, Motivation and Behaviour change model [33]. Health care provider (HCP) informed and educated the patients to enable self-care ability as well as parents' participation in the treatment plan. At a first 60 min individual session with one out of four experienced physiotherapists, patients were prescribed and instructed to perform adequate self-mediated physical activity of at least moderate intensity 60 min/daily [34]. A diary was distributed to enhance and monitor adherence to treatment. Reinforcement to the assigned intervention was performed at the clinic at follow-up every 6-months. Additional contact via telephone was used when needed.

**Night-time brace.** Additionally, the NB group received a custom-designed hypercorrective Boston scoliosis night brace [35] to be worn 8 hours every night. A senior orthotist (M.H.) approved the radiological in-brace corrections aiming for 50% improvement of curve magnitude for all patients in the study.

**Scoliosis-specific exercise.** Additionally, the SSE group were educated by a physiotherapist during the first session to perform exercises for self-correction. This includes spinal stability, muscular stabilisation, postural control, and endurance in corrective postures integrated in activities of daily living covering similar concepts and methods described in previous literature [36]. An initial dosage of 10 repetitions x 3 sets for 30 min was included in their prescription of 60 min self-mediated physical activity. Progression with skill transference to self-mediated activities as well as over-corrective side-shift were gradually introduced. Individual reinforcement of the intervention was conducted at the clinic by the same physiotherapist once per month for 60–90 min during the first 3 months.

**Adequate self-mediated physical activity as control group.** Patients in the PA group were prescribed adequate self-mediated physical activity of at least moderate intensity 60 min daily, throughout the study period [34].

### Data collection and self-reported outcome measures

Data was collected at inclusion and every 6 months. Descriptive data included age, gender, weight, height, body mass index [32], and self-report on maturity stages [37,38] for breast/genitals and pubic hair (Tanner). Clinical outcome measures included ATR assessed with

scoliometer (average of 2 measurements) [39] and radiographs. The Cobb angle [40] of major curve and ossification of the iliac apophysis (Risser) [41] were recorded from radiographs. Self-reported outcome measures included:

- The pictorial part of the Spinal Appearance Questionnaire (pSAQ) [42–45] captures the patient's perception of spinal shape asymmetry based on 7 categories (body curve, rib prominence, flank prominence, head–chest–hips relationship, head position over hips, shoulder level, and spine prominence). Each category is graded from 1 (best) to 5 (worst). The total score ranges from 7–35.

- The Scoliosis Research Society-22 revised (SRS-22r) questionnaire comprises 22 questions categorized into 5 domains (function, pain, self-image, mental health, and satisfaction) [46–48]. The subscore (function, pain, self-image, mental health), and total score (all five domains) are calculated. Each scale ranges from 1 (worst) to 5 (best). EuroQol 5-Dimensions Youth 3 Levels (EQ-5D-Y-3L) [49–54] comprises five domains (mobility, self-care, usual activities, pain or discomfort, and anxiety or depression). Each dimension has three levels of severity: from 1 (no) to 3 (a lot of) problems. Self-report on current health state is assessed with Visual-Analogue-Scale (EQ-VAS), from 0 (worst) to 100 (best).

- To gain knowledge of desired behavioural outcomes [33] regarding the treatment plan, patients were asked to report on 3 questions "The grade to which you feel that you have completed the treatment (adherence)", "The grade to which you are motivated to carry out the treatment (motivation)", and "How confident are you in your own capability to perform the treatment (capability)", from 1 (very high grade) to 4 (not at all). HCP´s reported 1 question on adherence using the diary in dialogue with the patient and family "To what grade the patient had adhered to treatment", from 1 (very sure) to 4 (not at all).

- International Physical Activity Questionnaire short form (IPAQ-SF) [55,56] was used to capture self-report on adherence to physical activity including vigorous, moderate, and walking activities during the last 7 days (at least 10 minutes). Each activity is multiplied with the metabolic equivalent of a task (MET). The MET values are 8.0 for vigorous intensity, 4.0 for moderate intensity, and 3.3 for walking [55]. The MET value is multiplied with number of days and minutes for each activity into minutes/week. Total MET-minutes/week were provided by a summation of all activities (vigorous, moderate, walking). The score ranges from 0 minutes/day to180 maximum minutes/day for each level and a maximum of 21 hours of activity level/week to avoid overestimation [56]. An additional question about sedentary behaviors (sitting) on an ordinary day is reported in minutes/day.

The endpoint for the study was curve progression of 6° or less at skeletal maturity (treatment success) or curve progression of more than 6° (treatment failure) seen on two consecutive posteroanterior standing radiographs compared to inclusion [29–31]. In case of treatment failure, either a full-time thoracolumbosacral Boston brace was offered, or surgery for cases where the scoliosis progressed to a Cobb angle of 45° or more with remaining growth. The current paper analyzes secondary outcome findings, including change in clinical and self-reported outcome measures from baseline to endpoint of treatment. The primary outcome findings have previously been reported by our research group concerning the probability of curve survival of 6° or less overtime for each treatment group [10].

## Sample size and randomization

Based on previous literature on the primary outcome, an estimation of 45 patients in each of the treatment groups were required. This to achieve a power of 80% to detect between-group

differences with a significance level of 5% [29]. The failure rate for the NB and SSE groups was set at 15%, and at 45% in the PA group [57] with an estimated dropout rate of up to 20%. In total, a sample size of 135 patients were required.

The HCP´s enrolled patients after the following procedures: informed written consent; central research coordinator contact for the randomization process regarding intervention allocation. The randomization was done with a 1:1:1 ratio into NB, SSE, or PA group (Fig 1). The randomization was based on a priori computer-generated random numbers table in varying block sizes prepared by an independent statistician. The same statistician prepared consecutively numbered, sealed opaque envelopes which were kept in a locked location and only opened in the presence of at least two persons. In this study, blinding of patients and HCP´s regarding interventions as well as outcome measures were not possible during the study. After the study completion, all radiographs were also reassessed independently by two blinded experienced spine surgeons on Cobb angle measurements.

## Statistical methods and analyses

Data was double-entered, and any discrepancies were corrected. Descriptive data were calculated as mean and standard deviation (±), 95% confidence interval (95% CI), or as number and proportions (no, %) as adequate.

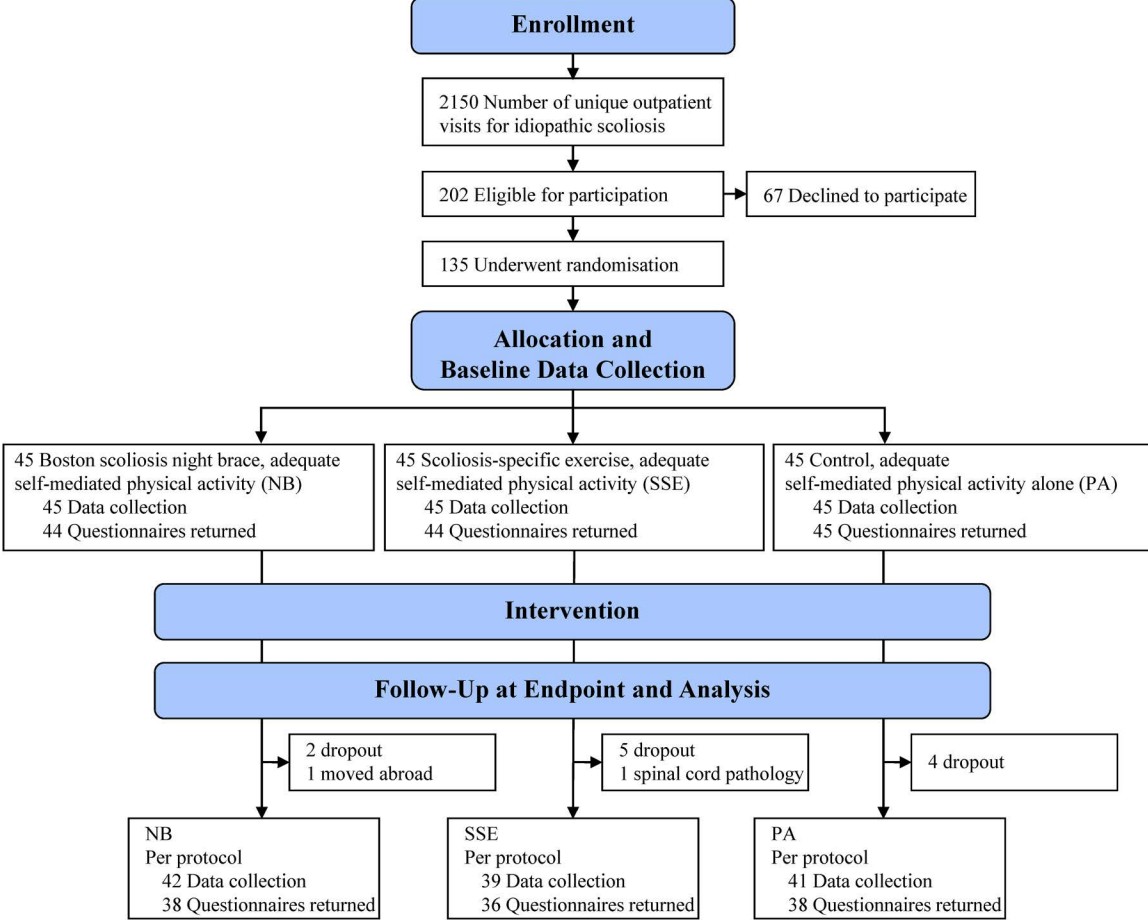

**Fig 1. CONSORT flow diagram of participants with AIS from enrollment to endpoint including PRO-specific extensions in Supporting information.**

Patient and HCP agreement in reporting of adherence was estimated with linear weighted kappa (K), and the relative strength of agreement according to poor <0.00, slight 0.00–0.20, fair 0.21–0.40, moderate 0.41–0.60, substantial 0.61–0.80, and almost perfect 0.81–1.00 [58]. Logistic regression assessed the association of motivation and capability (two independent factors) with adherence, applying dichotomized ratings (a very high grade & high grade compared to low grade & not at all). Nagelkerke pseudo R-squared ($R^2$) and odds ratio (OR) statistics are presented.

To explore possible within group and between-group differences and effects for continuous and categorical variables a significance level of $p < 0.05$ was used. Univariable Analysis of Variance for secondary outcomes at endpoint were assessed with partial eta squared ($\eta_{p^2}$) with an estimated measure of effect size [59] as; $\eta_{p^2} = 0.01$ ~ small effect, $\eta_{p^2} = 0.06$ ~ medium effect, and $\eta_{p^2} = > 0.14$ ~ large effect or Chi-Square test as appropriate. Linear Mixed Model analyses [60] for repeated measures with unstructured covariance were performed for secondary outcomes to investigate change between baseline to endpoint, with time and treatment group as fixed effects presented as an interaction term. Pairwise comparison was used with mean difference (95% CI) if main interaction effect was significant. Through application of restricted maximum likelihood imputation methods, intention-to-treat analyses were performed, i.e., regardless of adherence or loss to follow-up, all participants remained randomized in the analysis. The data set contained missing data ranging from 15–23% per variable with the exception of the IPAQ sitting min/day variable with 44% missing data.

Satisfaction domain and total score in SRS-22r and EQ-5D-Y-3L domains were analyzed per protocol including Paretian analysis with a classification of health change towards worse, unchanged, better, or indeterminable [61]. Furthermore, sensitivity analyses were performed in those with complete data per protocol and also regarding Complier Average Causal Effect (CACE) [62] with at least 50% HCP-reported patient adherence to the treatment protocol at 6-monthly intervals until endpoint of study. Additional sensitivity analyses were performed on subgroups whose Cobb angle progressed more than 6° and those reaching maturity without reaching threshold for progression, and those with Risser stage 2 or less and those with Risser stage 3 or more. Statistical Package for the Social Sciences (SPSS V29.0.0.0, IBM Corporation, New York, NY, USA) statistical software for Windows and Mplus (version 8.2) were used.

## Results

### Participant flow and baseline characteristics

Fig 1 displays patient flow [63] throughout the study where 135 patients were successfully randomized. Out of 135 allocated patients, 122 patients (90%) remained in the study until endpoint (Fig 1). Baseline characteristics (Table 1), mean age 12.7 (±1.4) years, and mean Cobb angle 31° (±4.3) showed no statistically significant differences between the treatment groups. The mean time (SD) in study until endpoint was in the NB group 22.8 (±12.9) months while the SSE and PA groups remained in study during 16.1 (±10.6), and 16.2 (±10.5) months respectively.

### Angle of trunk rotation and cobb angle of major curve

ATR showed a statistically significant increase in within-group mean change from baseline to endpoint for the SSE group with 2.1° (95% CI = 0.9 to 3.3), and 1.6° for the PA group (95% CI = 0.5 to 2.7). A statistically significant between-group difference in change between baseline to endpoint in main interaction effect was displayed favouring the NB group with - 2.0° (95% CI = -3.7 to -0.3) compared to the SSE group (Table 2).

**Table 1. Baseline characteristics of the included patients.**

| | Overall sample N=135 | NB N=45 | SSE N=45 | PA N=45 |
|---|---|---|---|---|
| Age, years[a] | 12.7 (1.4) | 12.7 (1.4) | 12.6 (1.4) | 12.6 (1.5) |
| Females – n (%) | 111 (82) | 39 (87) | 33 (73) | 39 (87) |
| Height (cm)[a] | 158.1 (9.5) | 157.2 (9.5) | 158.1 (9.6) | 159.0 (9.6) |
| Weight (kg)[a] | 46.0 (9.2) | 44.8 (9.3) | 45.7 (9.0) | 47.3 (9.4) |
| Body Mass Index[a] | 18.3 (2.6) | 18.0 (2.7) | 18.2 (2.4) | 18.6 (2.7) |
| Tanner scale, breast/genitals – n (%)[b] | 125 (93) | 42 (93) | 41 (91) | 42 (93) |
| Stage I | 8 (6) | 2 (5) | 5 (12) | 1 (2) |
| Stage II | 23 (18) | 8 (19) | 5 (12) | 10 (24) |
| Stage III | 65 (52) | 22 (52) | 20 (49) | 23 (55) |
| Stage IV | 26 (21) | 10 (24) | 9 (22) | 7 (17) |
| Stage V | 3 (2) | 0 (0) | 2 (5) | 1 (2) |
| Tanner scale, pubic hair – n (%)[b] | 123 (91) | 41 (91) | 40 (89) | 42 (93) |
| Stage I | 16 (13) | 5 (12) | 6 (15) | 5 (12) |
| Stage II | 19 (15) | 6 (15) | 6 (15) | 7 (17) |
| Stage III | 33 (27) | 14 (34) | 9 (22) | 10 (24) |
| Stage IV | 49 (40) | 14 (34) | 15 (38) | 20 (48) |
| Stage V | 6 (5) | 2 (5) | 4 (10) | 0 (0) |
| Risser stage – n (%)[c] | 128 (95) | 42 (93) | 42 (93) | 44 (98) |
| 0 | 68 (53) | 21 (50) | 22 (52) | 25 (57) |
| 1 | 14 (11) | 5 (12) | 4 (10) | 5 (11) |
| 2 | 14 (11) | 5 (12) | 3 (7) | 6 (14) |
| 3 | 23 (18) | 9 (21) | 8 (19) | 6 (14) |
| 4 | 9 (7) | 2 (5) | 5 (12) | 2 (4) |
| 5[d] | | | | |
| Angle of trunk rotation (degrees)[a] | 11.3 (3.1) | 11.8 (2.7) | 10.8 (3.3) | 11.3 (3.1) |
| Cobb angle, major curve (degrees)[a] | 31 (4.3) | 32 (4.5) | 31 (4.1) | 31 (4.4) |
| Major Curve location – n (%) | | | | |
| Thoracic | 92 (68) | 31 (69) | 31 (69) | 30 (67) |
| Thoracolumbar | 27 (20) | 9 (20) | 7 (16) | 11 (24) |
| Lumbar | 16 (12) | 5 (11) | 7 (16) | 4 (9) |

[a]Values are mean and standard deviation.

[b]Self-report on sexual maturity.

[c]Skeletal maturity assessment of ossification of iliac apophysis.

[d]Zero patients with Risser Stage 5.

Body Mass index: Weight in kilograms divided by the square of the height in meters; N: Number of patients; NB: Boston scoliosis night brace; SSE: scoliosis-specific exercise; PA: control with adequate self-mediated physical activity alone.

The within-group mean change for the Cobb angle showed a statistically significant increase from baseline to endpoint for all groups. The NB group showed an increase of 4.9° (95% CI =2.7 to 7.2), the SSE group 6.8° (95% CI = 4.5 to 9.1), and the PA group 7.9° (95% CI =5.6 to 10.2) respectively. No statistically significant between-group differences in change between baseline to endpoint in main interaction effect were displayed (Table 2).

## Perception of spinal appearance

Self-report on pSAQ displayed a statistically significant negative within-group mean change in the SSE group 1.6 (95% CI = 0.4 to 2.9) between baseline and endpoint. No statistically

**Table 2. Angle of trunk rotation, spinal deformity and appearance, within-group and between-group differences in change between baseline to endpoint.**

| | Baseline | Within-group change between baseline to endpoint | | Between-group differences (NB-SSE; SSE-PA; NB-PA) in change between baseline to endpoint | |
|---|---|---|---|---|---|
| | | Mean change | | F; | Pairwise comparison |
| | Mean (CI) | (CI) | p-value | p-value [e] | Mean difference (CI); p-value |
| | | Positive change = worse outcome | | | Favored group ↑ |
| Angle of trunk rotation (degrees) | | | | 3.2; **0.045** | |
| NB | 11.8 (10.9 to 12.7) | < 0.1 (-1.1 to 1.2) | 0.914 | | NB↑-SSE= -2.0 (-3.7 to -0.3); **0.019** |
| SSE | 10.7 (9.8 to 11.6) | 2.1 (0.9 to 3.3) | **< 0.001** | | SSE-PA= 0.5 (-1.2 to 2.1); 0.565 |
| PA | 11.3 (10.4 to 12.2) | 1.6 (0.5 to 2.7) | **0.006** | | NB-PA= -1.5 (-3.2 to 0.1); 0.063 |
| Cobb angle (degrees) | | | | 1.7; 0.186 | |
| NB | 32.0 (30.7 to 33.3) | 4.9 (2.7 to 7.2) | **< 0.001** | | NA |
| SSE | 31.1 (29.8 to 32.4) | 6.8 (4.5 to 9.1) | **< 0.001** | | |
| PA | 31.2 (29.9 to 32.5) | 7.9 (5.6 to 10.2) | **< 0.001** | | |
| pSAQ (7–35) | | | | 1.0; 0.358 | |
| NB | 11.8 (10.9 to 12.8) | 0.4 (-0.8 to 1.6) | 0.484 | | NA |
| SSE | 11.3 (10.4 to 12.3) | 1.6 (0.4 to 2.9) | **0.009** | | |
| PA | 11.7 (10.8 to 12.7) | 0.8 (-0.4 to 2.0) | 0.168 | | |

NB: Boston scoliosis night brace; SSE: scoliosis-specific exercise; PA: control with adequate self-mediated physical activity alone; pSAQ: pictorial Spinal Appearance Questionnaire; ↑: Favored group marked with an arrow; CI: 95% confidence interval; Bold: significant at p < 0.05;

[e]time and treatment group presented as an interaction term; NA: not applicable.

significant between-group differences in main interaction effect from baseline to endpoint were displayed (Table 2).

## Health-related quality of life

Table 3 presents results from the SRS-22r. The NB group had a within-group mean change showing a statistically significant but slight decrease in function from baseline to endpoint (-0.1, 95% CI = -0.3 to <-0.1). The within-group analysis for the SSE group displayed a statistically significant slight worsening in three domains with increase of pain (-0.2, 95% CI = -0.3 to <-0.1), self-image (-0.3, 95% CI = -0.5 to -0.1), mental health (-0.3, 95% CI = -0.5 to <-0.1) and subsequently lower subscore (-0.2, 95% CI = -0.3 to -0.1). No statistically significant between-group differences in change between baseline to endpoint in main interaction effect were displayed (Table 3). Likewise, the satisfaction domain and total score displayed no differences in main effects between groups at endpoint (Table 3).

Also outlined in Table 3, EQ-VAS health state shows no differences for within-group change nor between-group differences in change between baseline to endpoint in main interaction effect. But, in the EQ-5D-Y-3L dimensions (Table 4) there was a statistically significant difference between groups regarding mobility (p = 0.031) and usual activities (p = 0.003) at endpoint with the SSE group reporting higher proportion of moderate problems compared to the NB and PA groups. However, the Paretian Classification of Health Change showed no statistically significant difference between groups from baseline to endpoint (Table 4).

## Adherence, motivation, and capability

Table 5 displays that in most cases a "high grade" of adherence was reported by patients whereas HCP´s in most cases reported "very sure" regarding patient adherence at endpoint. Furthermore, the NB and PA groups reported a larger proportion of "high grade" compared to the SSE group (p = 0.012). Likewise, the SSE group reported a higher proportion "low grade"

**Table 3. SRS-22r and EQ-VAS health state, within-group change and between-group differences between baseline to endpoint.**

| | Baseline | Within-group change between baseline to endpoint | | Between-group differences (NB-SSE; SSE-PA; NB-PA) in change between baseline to endpoint | |
| --- | --- | --- | --- | --- | --- |
| | | | | F; p-value [e] | Pairwise comparison |
| | Mean (CI) | Mean change (CI) | p-value | | Mean difference (CI); p-value |
| SRS-22r | | Negative change = worse outcome | | | Favored group < |
| Function (1–5) | | | | 0.7; 0.505 | |
| NB | 4.8 (4.7 to 4.9) | -0.1 (-0.3 to <-0.1) | **0.018** | | NA |
| SSE | 4.7 (4.6 to 4.8) | -0.1 (-0.2 to < 0.1) | 0.059 | | |
| PA | 4.7 (4.7 to 4.8) | <-0.1 (-0.2 to 0.1) | 0.424 | | |
| Pain (1–5) | | | | 1.1; 0.351 | |
| NB | 4.7 (4.5 to 4.9) | <-0.1 (-0.2 to 0.1) | 0.480 | | NA |
| SSE | 4.6 (4.4 to 4.8) | -0.2 (-0.3 to <-0.1) | **0.028** | | |
| PA | 4.7 (4.5 to 4.9) | <-0.1 (-0.2 to 0.1) | 0.765 | | |
| Self-image (15) | | | | 2.2; 0.110 | |
| NB | 4.1 (3.9 to 4.3) | < 0.1 (-0.2 to 0.2) | 0.875 | | NA |
| SSE | 4.2 (4.0 to 4.3) | -0.3 (-0.5 to -0.1) | **0.012** | | |
| PA | 4.2 (4.0 to 4.4) | <-0.1 (-0.2 to 0.2) | 0.858 | | |
| Mental health (1–5) | | | | 0.2; 0.808 | |
| NB | 4.3 (4.2 to 4.5) | - 0.2 (-0.4 to < 0.1) | 0.121 | | NA |
| SSE | 4.3 (4.1 to 4.5) | - 0.3 (-0.5 to <-0.1) | **0.020** | | |
| PA | 4.3 (4.1 to 4.5) | - 0.2 (-0.4 to < 0.1) | 0.104 | | |
| Subscore (1–5) | | | | 1.6; 0.210 | |
| NB | 4.5 (4.4 to 4.6) | -0.1 (-0.2 to < 0.1) | 0.154 | | NA |
| SSE | 4.5 (4.3 to 4.6) | -0.2 (-0.3 to -0.1) | **< 0.001** | | |
| PA | 4.5 (4.4 to 4.6) | -0.1 (-0.2 to < 0.1) | 0.106 | | |
| | | | Between-group differences (NB-SSE; SSE-PA; NB-PA) at endpoint | | |
| | | | | | Pairwise comparison |
| At endpoint only | Mean (CI) | | Main effects F; p-value; $\eta_p^2$ | | Mean difference (CI); p-value |
| Satisfaction (1–5) | | | 0.5; 0.600; 0.009 | | |
| NB | 3.7 (3.4 to 4.0) | | | | NA |
| SSE | 3.5 (3.2 to 3.8) | | | | |
| PA | 3.7 (3.4 to 4.0) | | | | |
| Total score (1–5) | | | 1.5; 0.217; 0.028 | | |
| NB | 4.3 (4.2 to 4.5) | | | | NA |
| SSE | 4.2 (4.0 to 4.4) | | | | |
| PA | 4.4 (4.2 to 4.5) | | | | |
| | | Within-group change from baseline to endpoint | | Between-group differences (NB-SSE; SSE-PA; NB-PA) in change between baseline to endpoint | |
| | Baseline | | | | Pairwise comparison |
| EQ-VAS | Mean (CI)) | Mean change (CI) | p-value | F; p-value [e] | Mean difference (CI); p-value |
| Health state (0–100) | | | | 0.4; 0.684 | |
| NB | 88.6 (85.5 to 91.8) | -1.6 (-5.5 to 2.2) | 0.410 | | NA |
| SSE | 87.8 (84.7 to 91.0) | -3.8 (-7.8 to 0.1) | 0.059 | | |
| PA | 86.8 (83.7 to 90.0) | -3.6 (-7.5 to 0.3) | 0.070 | | |

SRS-22r: The Scoliosis Research Society-22r (1–5 best); EQ-VAS: EuroQol Visual-Analogue-Scale (0–100 best); NB: Boston scoliosis night brace; SSE: scoliosis-specific exercise; PA: control with adequate self-mediated physical activity alone CI: 95% confidence interval; Bold: significant at p < 0.05; $\eta_p^2$: partial eta squared; [e] time and treatment group presented as an interaction term; NA: not applicable.

**Table 4. EQ-5D-Y-3L at baseline and endpoint, and Paretian classification of health change from baseline to endpoint.**

| EQ-5D-Y-3L | Mobility | | Self-care | | Usual activities | | Pain/discomfort | | Anxiety/depression | |
|---|---|---|---|---|---|---|---|---|---|---|
| | Baseline | Endpoint | Baseline | Endpoint | Baseline | Endpoint | Baseline | Endpoint | Baseline | Endpoint |
| | (N=133) | (N=112) | (N=133) | (N=111) | (N=133) | (N=112) | (N=133) | (N=112) | (N=130) | (N=111) |
| **No problems** | | | | | | | | | | |
| NB - number (%) | 44 (100) | 38 (100) | 44 (100) | 37 (100) | 42 (96) | 37 (97) | 34 (77) | 29 (76) | 28 (65) | 26 (70) |
| SSE | 43 (98) | 33 (92) | 44 (100) | 36 (100) | 41 (93) | 31 (86) | 33 (75) | 21 (58) | 31 (74) | 26 (72) |
| PA | 45 (100) | 38 (100) | 45 (100) | 38 (100) | 43 (96) | 38 (100) | 34 (76) | 29 (76) | 32 (71) | 28 (74) |
| **Moderate problems** | | | | | | | | | | |
| NB - number (%) | 0 (0) | 0 (0) | 0 (0) | 0 (0) | 2 (4) | 0 (0) | 9 (20) | 8 (21) | 14 (33) | 10 (27) |
| SSE | 1 (2) | **3 (8)** | 0 (0) | 0 (0) | 3 (7) | **5 (14)** | 9 (20) | 14 (39) | 11 (26) | 9 (25) |
| PA | 0 (0) | 0 (0) | 0 (0) | 0 (0) | 2 (4) | 0 (0) | 10 (22) | 9 (24) | 12 (27) | 7 (18) |
| **Severe problems** | | | | | | | | | | |
| NB - number (%) | 0 (0) | 0 (0) | 0 (0) | 0 (0) | 0 (0) | 1 (3) | 1 (2) | 1 (3) | 1 (2) | 1 (3) |
| SSE | 0 (0) | 0 (0) | 0 (0) | 0 (0) | 0 (0) | 0 (0) | 2 (4) | 1 (3) | 0 (0) | 1 (3) |
| PA | 0 (0) | 0 (0) | 0 (0) | 0 (0) | 0 (0) | 0 (0) | 1 (3) | 0 (0) | 1 (2) | 3 (8) |
| p-value | 0.662 | **0.031** | NA | NA | 0.898 | **0.003** | >0.999 | 0.249 | 0.852 | 0.752 |

**Paretian Classification of Health Change, between baseline to endpoint (N=109)**

| | Worse[1] | Unchanged[2] | Better[3] | Indeterminable[4] | p-value |
|---|---|---|---|---|---|
| NB n=37 - number (%) | 7 (19) | 23 (62) | 5 (14) | 2 (5) | |
| SSE n=34 | 11 (32) | 17 (50) | 5 (15) | 1 (3) | 0.727 |
| PA n=38 | 8 (21) | 20 (53) | 9 (24) | 1 (3) | |

N: number of patients; EQ-5D-Y-3L: EuroQol 5-Dimensions Youth 3 Levels; NB: Boston scoliosis night brace; SSE: scoliosis-specific exercise; PA: control with adequate self-mediated physical activity alone; NA: not applicable, the distribution is a constant in self-care;

[1]: the health state is worse in at least one dimension, and is no better in any other dimension;

[2]: the health state is exactly the same;

[3]: the health state is better in at least one dimension and is no worse in any other dimension;

[4]: the changes in health are "mixed", better in one dimension, but worse in another; Bold: significant at p < 0.05.

**Table 5. Adherence, motivation, capability, and agreement in reporting of adherence to treatment at endpoint.**

| Self-reported (N=110) | NB | SSE | PA | p-value |
|---|---|---|---|---|
| | (N=38) | (N=36) | (N=36) | |
| Adherence - number (%) | | | | **0.012** |
| Very high grade | 8 (21) | 7 (19) | 8 (22) | |
| High grade | **24** (63) [a] | **11** (31) [b] | **22** (61) [a] | |
| Low grade | **5** (13) [b] | **16** (44) [a] | **6** (17) [b] | |
| Not at all | 1 (3) | 2 (6) | 0 (0) | |
| Motivation - number (%) | | | | 0.221 |
| Very high grade | 10 (27) | 9 (25) | 9 (25) | |
| High grade | 11 (30) | 14 (39) | 20 (56) | |
| Low grade | 12 (32) | 11 (31) | 7 (19) | |
| Not at all | 4 (11) | 2 (6) | 0 (0) | |
| Capability - number (%) | | | | 0.529 |
| Very high grade | 17 (47) | 12 (33) | 18 (50) | |
| High grade | 12 (33) | 17 (47) | 15 (42) | |
| Low grade | 6 (17) | 5 (14) | 3 (8) | |
| Not at all | 1 (3) | 2 (6) | 0 (0) | |
| HCP-reported on patient adherence (N=115) | (N=39) | (N=37) | (N=39) | |
| Adherence - number (%) | | | | **0.013** |
| Very sure | 19 (49) | **10** (27) [b] | **23** (59) [a] | |
| Sure | 5 (13) | 11 (30) | 10 (26) | |
| Unsure | 8 (20) | **13** (35) [a] | **4** (10) [b] | |
| Not at all | 7 (18) | 3 (8) | 2 (5) | |

| Self-report-HCP-report, Agreement in reporting of adherence to treatment (N=107) | | | | |
|---|---|---|---|---|
| | HCP-reported patient adherence | | | |
| | Very sure | Sure | Unsure | Not at all |
| Self-reported adherence | | | | |
| Very high grade | 19 | 2 | 2 | 0 |
| High grade | 25 | 20 | 6 | 3 |
| Low grade | 5 | 3 | 14 | 5 |
| Not at all | 0 | 0 | 0 | 3 |
| | Linear weighted Kappa = 0.439 | | | |

N: number; NB: Boston scoliosis night brace; SSE: scoliosis-specific exercise; PA: control with adequate self-mediated physical activity alone; HCP: health care provider; Bold: statistical significance at p < 0.05;

[a,b]: where row proportion a > b.

of adherence regarding the treatment plan compared to the NB and PA groups (p = 0.012). The HCP´s reported that the PA group had a larger proportion of patients being "very sure" regarding adherence compared to the SSE group (p = 0.013). The SSE group also had a larger proportion of HCP´s reporting "unsure" compared to the PA group (p = 0.013). However, patient report on motivation and capability showed no statistically significant between-group differences. The strength of patient-HCP proportional agreement was moderate (K = 0.439) in reporting adherence to the treatment plan. Patient-HCP proportional agreement on treatment adherence most frequently occurred within the "high grade-sure" (n = 20/107), and the "very high grade-very sure" (n = 19/107) answers. Whereas the largest mismatch appeared between "high grade-very sure (n =25/107) answers.

Regardless of group allocation the variation in adherence at endpoint showed that 53% was explained by motivation and capability ($R^2$ = 0.53) in logistic regression. Patients with high self-reported motivation had 11.1 times better odds of high adherence (OR = 11.12, 95% CI = 2.9 to 43.3). Comparably, patients with high self-reported capability had 7.2 times better odds of high adherence to the treatment plan (OR = 7.23, 95% CI = 1.5 to 34.4).

## Physical activity

Table 6 presents results of the IPAQ-SF. The within-group change between baseline to endpoint, showed a statistically significant positive increase for the SSE group in walking 142 minutes/week (p = 0.011). There were no statistically significant between-group differences in change between baseline to endpoint in main interaction effect.

**Table 6. Levels of physical activity and sedentary behaviour according to IPAQ-SF, within-group and between-group differences in change between baseline to endpoint.**

| IPAQ-SF | Baseline Mean (CI)) | Within-group change between baseline to endpoint Mean change (CI) | p-value | Between-group differences in change between baseline to endpoint (NB-SSE; SSE-PA; NB-PA) F; p-value [e] | Pairwise comparison Mean difference (CI); p-value |
|---|---|---|---|---|---|
| Levels of Physical Activity | | Positive change= better outcome | | | |
| Vigorous-min/week | | | | 1.5; 0.224 | |
| NB | 168 (98–238) | -30 (-99–39) | 0.397 | | NA |
| SSE | 156 (87–226) | 13 (-57–84) | 0.707 | | |
| PA | 116 (47–185) | 56 (-13–126) | 0.110 | | |
| Moderate-min/week | | | | 0.3; 0.727 | |
| NB | 174 (102–247) | 16 (-76–108) | 0.734 | | NA |
| SSE | 181 (107–255) | 67 (-27–161) | 0.158 | | |
| PA | 146 (73–220) | 52 (-42–145) | 0.274 | | |
| Walking-min/week | | | | 2.9; 0.059 | |
| NB | 314 (228–401) | -43 (-149–63) | 0.425 | | NA |
| SSE | 213 (126–301) | 142 (33–252) | **0.011** | | |
| PA | 213 (126–299) | 48 (-60–155) | 0.379 | | |
| Sedentary behaviour | | Negative change= better outcome | | | |
| Sitting-min/day | | | | 2.1; 0.131 | |
| NB | 434 (376–493) | -45 (-113–23) | 0.188 | | NA |
| SSE | 390 (331–450) | 45 (-20–111) | 0.169 | | |
| PA | 419 (362–476) | -25 (-87–36) | 0.416 | | |
| Total MET-min/week | | Positive change= better outcome | | | |
| | | | | 2.1; 0.123 | |
| NB | 3025 (2233–3815) | -244 (-1104–616) | 0.576 | | NA |
| SSE | 2698 (1890–3505) | 885 (-3–1774) | 0.051 | | |
| PA | 2202 (1404–3000) | 839 (-29–1707) | 0.058 | | |

IPAQ-SF: International Physical Activity Questionnaire-Short Form; MET-min/week: Total metabolic equivalent of task for vigorous, moderate and walking activity minutes/week; NB: Boston scoliosis night brace; SSE: scoliosis-specific exercise; PA: control with adequate self-mediated physical activity alone; CI: 95% confidence interval; Bold: significant at p < 0.05;

[e] time and treatment group presented as an interaction term; NA: not applicable.

### Sensitivity and subgroup analyses

Sensitivity analysis applying the CACE method showed similar results for between-group comparisons of all outcome measures. However, the few between-group differences found did not remain when analyzing those with complete data per protocol or in subgroup sensitivity analyses. Patient characteristics were similar for those who completed patient questionaries at endpoint compared to non-responders.

## Discussion

Our study showed only few between-group differences in change from baseline to endpoint in outcome measures for either NB, SSE, or PA treatments for AIS. One of these, was the better effect of NB compared to SSE in preventing progression of ATR. This difference is likely greater than error of measurement but does not exceed the intra-rater smallest real difference reported in earlier literature of approximately 4° [64]. The SSE group also showed a small but significantly worse outcome in pSAQ, as well as SRS-22r self-image, pain and mental health domains but not likely reaching clinical relevance [65,66]. The SSE and PA groups had a within-group progression in ATR while the Cobb angle showed a progression for all groups. For the NB group, the progression in Cobb angle was on average under 6° and above for the PA and SSE groups, but this did not eventuate in significant between-group differences in Linear Mixed Models. SSE compared to the other groups showed an increase of moderate problems in mobility and usual activity dimensions of the EQ-5D-Y-3L. But health state and health change did not differ between groups from baseline to endpoint according to EQ-VAS and Paretian Classification. These results contrast with literature from heterogeneous scoliosis-specific exercise interventions suggesting improvement in HRQoL outcomes [67]. One can however interpret that in our study, SSE for moderate-grade AIS is not effective in preventing worse outcome in ATR, self-image, mental health and usual activities compared to NB and PA. This possibly due to greater patient focus on awareness of spinal and trunk deformity.

Our recent report [10] on primary outcome showed superiority for the NB group compared to the PA group but non-superiority for the SSE group compared to PA group. The current analyses on secondary outcomes show that the NB group had a significant within-group decrease in function according to SRS-22r but still on a high level with no decrease in any other HRQoL outcomes. In previous literature, full-time bracing has shown in some studies a moderate sized negative impact on different aspects of HRQoL [68,69] while another study showed no negative impact [70]. Our study however showed no negative impact on health when comparing the NB group with SSE and PA groups.

Another between-group difference was report on adherence at endpoint. Adherence was generally higher in the NB and PA groups compared to the SSE group. However, patient report on motivation and capability to perform treatment showed no differences between-groups, and variation in adherence was explained by as much as 53% of motivation and capability. It was also evident that the importance of motivation with regards to long-term treatment adherence increased, whereas capability was of more importance during the first 6 months of intervention [71]. Despite having good motivation and capability, the complexity of the intervention and contextual factors may influence one´s opportunity to adhere to the intervention. This was potentially more evident in the SSE intervention where learning new motor skills, integrating them into daily life (dual task), and maintaining a behaviour change is a challenge [72–75]. Night-time bracing may be considered a simpler intervention, and current literature suggests good compliance due to no restrictions during daytime activities [1,7,9].

The current study was performed in the Swedish public health care systems´ outpatient setting which is generalizable for all scoliosis health care in Sweden and to similar health care settings internationally. Our study had a pre-published a priori protocol with rigorous design, concealed randomization and allocation by independent statistician. The a priori sample size calculation was however based on the primary outcome with resulting data collection maintaining adequate power with consideration to dropouts and treatment failure. A posteriori statistical power assessment also suggested adequate power for secondary outcomes. Blinding of patients and HCP´s regarding interventions and outcome measures were not possible during the study. Moreover, double entry with mismatch control of all clinical measurements and reported outcomes was performed before data analyses. Another strength is the use of ITT, CACE and sensitivity analyses showing similar results.

SSE in the current study covers similar concepts and methods to those described in previous literature on scoliosis specific treatments [11,36]. However, our study's main focus was on enabling and supporting patient self-management in the outpatient setting throughout the long-term intervention. Acceptable adherence to the treatment plan, and good retention in prospectively collected data provided low risk of bias compared to previous studies. Also, the use of self-reported outcome and experience measurements to capture the patient perspective regarding the allocated treatment was a strength in the study.

A limitation of this study is that the report on adherence, motivation, and capability are based on the treatment plan as a whole. Low adherence towards one part of the treatment plan can therefore not be disregarded. However, IPAQ showed that all groups achieved adequate levels of self-mediated physical activity at baseline and endpoint [56]. This study only comprises subjective reporting of adherence, including patient and HCP-administered questionnaires. However, the moderate level of agreement between patient and HCP rated data provide an adequate level of validity to our treatment adherence analysis. Measuring adherence is complex, and literature suggests an overestimation of 10% for self-reported brace wear compared to temperature sensors [76]. Antoine et al. [77] used temperature sensors and found that the first week is crucial for accepting night-time bracing and that the hypercorrection does not influence compliance negatively. Objective monitoring of physical activity still shows a broad diversity of instruments, and no established best practice standard exists [78]. Review of diaries at follow-up was a strategy to maintain and report adherence as accurately as possible, but patient adherence with the use of a diary varied in quality. The current study was planned in 2011–2012 [29] when the availability of objective and patient-friendly technologies for monitoring adherence was not available in our multi-center setting.

In our study, 18% of the population had a Risser stage 3 and 7% had Risser stage 4. One may consider that these patients had less risk for progression of their scoliosis, but literature suggests limited sensitivity of Risser stage 2 cut-off as 25% of the patients have significant remaining growth [79]. Furthermore, patients in current study still had remaining growth for at least one year [32] with clear indication for treatment, and randomization produced even distribution making the intervention groups comparable. Sensitivity analyses were also performed based on Risser subgrouping showing no difference in between group results.

## Conclusions

Night-time brace, scoliosis-specific exercise or physical activity interventions for adolescents with idiopathic scoliosis showed small differences between groups in trunk rotation, spinal deformity and appearance, health-related quality of life, and treatment adherence but not likely reaching clinical relevance.

## Supporting information

**S1 File. Detailed description of interventions.**
(DOCX)

**S2 File. CONSORT 2010 statement checklist item with PRO-specific extensions.**
(DOCX)

**S3 File. Ethics documentation CONTRAIS Swedish English.**
(PDF)

**S4 File. PLOSOne human subjects research checklist.**
(DOCX)

**S5 File. Response to the requested edits.**
(DOCX)

**S6 File. The CONTRAIS Study Group.**
(DOCX)

**S7 File. Data set.**
(PDF)

## Acknowledgments

We are grateful to all the patients for participating in this study. The authors give thanks to research coordinators Luigi and MariaWikzén (research nurse and study coordinator), physiotherapists Sofia Berto and Suzanne Thorén, as well as to Peter Endler, Panayiotis Savvides, Tomas Reigo, Björn Dahlman, and Ingrid Ekenman for patient recruitment. We also give thanks to Henrik Hedevik for statistical analysis.

## Author contributions

**Conceptualization:** Marlene Dufvenberg, Hans Möller, Paul Gerdhem, Allan Abbott.

**Data curation:** Marlene Dufvenberg, Anastasios Charalampidis, Allan Abbott.

**Formal analysis:** Marlene Dufvenberg, Allan Abbott.

**Funding acquisition:** Paul Gerdhem, Allan Abbott.

**Investigation:** Anastasios Charalampidis, Elias Diarbakerli, Hans Tropp, Anna Aspberg Ahl, Hans Möller, Paul Gerdhem, Allan Abbott.

**Methodology:** Marlene Dufvenberg, Birgitta Öberg, Allan Abbott.

**Project administration:** Allan Abbott.

**Resources:** Paul Gerdhem, Allan Abbott.

**Software:** Allan Abbott.

**Supervision:** Birgitta Öberg, Hans Tropp, Paul Gerdhem, Allan Abbott.

**Validation:** Allan Abbott.

**Visualization:** Marlene Dufvenberg, Allan Abbott.

**Writing – original draft:** Marlene Dufvenberg, Allan Abbott.

**Writing – review & editing:** Marlene Dufvenberg, Allan Abbott.

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
