## [Decision Letter · Decision Letter 0]

2 Jan 2025

PONE-D-24-52488Trunk rotation, Spinal Deformity and Appearance, Health Related Quality of Life, and Treatment Adherence: Secondary Outcomes in a Randomized Controlled Trial on Conservative Treatment for Adolescent Idiopathic ScoliosisPLOS ONE

Dear Dr. Marlene Dufvenberg, Thank you for submitting your manuscript to PLOS ONE. After careful consideration, we feel that it has merit but does not fully meet PLOS ONE’s publication criteria as it currently stands. Therefore, we invite you to submit a revised version of the manuscript that addresses the points raised during the review process.

We look forward to receiving your revised manuscript.

Kind regards,

Taher Babaee

Academic Editor

PLOS ONE

Journal Requirements:

2. In the online submission form, you indicated that [Deidentified data will be available for sharing by the corresponding author upon reasonable request.]. 

3. One of the noted authors is a group or consortium [The CONTRAIS Study Group]. In addition to naming the author group, please list the individual authors and affiliations within this group in the acknowledgments section of your manuscript. Please also indicate clearly a lead author for this group along with a contact email address.

Reviewers' comments:

Reviewer's Responses to Questions

**Comments to the Author**

1. Is the manuscript technically sound, and do the data support the conclusions?

Reviewer #1: Yes

Reviewer #2: Yes

2. Has the statistical analysis been performed appropriately and rigorously?

Reviewer #1: Yes

Reviewer #2: Yes

3. Have the authors made all data underlying the findings in their manuscript fully available?

Reviewer #1: Yes

Reviewer #2: Yes

4. Is the manuscript presented in an intelligible fashion and written in standard English?

Reviewer #1: No

Reviewer #2: Yes

5. Review Comments to the Author

Reviewer #1: As the statistical reviewer I will focus on methods and reporting. The analytical approaches are approprriate.

Major

1) It was not clear what the outcome is until the statistical analysis section. All the outcomes are listed with the primary outcome mentioned as not being reported in this work. I suggest have a section of outcomes that are evaluated in this work first, and then a second section of outcomes not used in this study.

2) The power calculations are around the primary outcome, as is expected, but this is not an analysis of the primary outcome. Also the power calculations are around a massive effect (15% vs 45% failure in two of the groups) so anything smaller than that, this RCT will be underpowered to detect. So, overall, power is a major concern for this study, and findings need to be evaluated very concervatively. In particular, the interaction effect analyses will be grossly underpowered.

3) What do the univariable (not univariate) analyses add? I suggest dropping or moving to an appendix to simplify the paper.

3) there is no information on missing data. why wasn't multiple imputation used as a main analytical approach? or were all data complete? please clarify in the methods section as per the CONSORT statement.

Reviewer #2: Thank you for writing the valuable article. Following I suggested some corrections on your article which hopefully make it better for your audiences.

Introduction section

Line 75: The verb “aims” came for “treatments”. If you want to say many kinds of treatment, you should use treatments with the related verb aim. Otherwise, you should use treatment with the verb aims.

Conclusion section

Please describe better what results can derive from your study. You mentioned a general description of the study. It is recommended to describe more detail and don not use the abbreviation to better conclude the message of your study.

6. PLOS authors have the option to publish the peer review history of their article (what does this mean? ). If published, this will include your full peer review and any attached files.

**Do you want your identity to be public for this peer review?** For information about this choice, including consent withdrawal, please see our Privacy Policy .

Reviewer #1: No

Reviewer #2: No

---

## [Author Response · Author response to Decision Letter 1]

11 Feb 2025

Dear Academic Editor Taher Babaee

Thank you for the opportunity to thoroughly revise our manuscript. We are grateful for the reviewers' precise and constructive suggestions in their evaluation. We have submitted a revised manuscript with track changes and provided responses to each point raised by the academic editor and reviewers.

Journal Requirements

1. Please ensure that your manuscript meets PLOS ONE's style requirement

Author response:

Additional copyediting has now been performed and is saved in revised manuscript with track changes.

Author response:

Data set includes minimal anonymized data set necessary to replicate our study

-DB Table 2 Data document

-DB Table 3 Data document

-DB Table 4 Data document

-DB Table 5 Data document

-DB Table 6 Data document

-Table 2 Syntax

-Table 3 Syntax

-Table 4 Syntax

-Table 5 Syntax

-Table 6 Syntax

Detailed data, including personal data that directly or indirectly identifies patients, can only be provided upon reasonable request, provided that Swedish ethical approval has been obtained for such data. For this, we can refer to the following link. https://etikprovningsmyndigheten.se/wp-content/uploads/2024/05/Guide-to-the-ethical-review_webb.pdf

3. One of the noted authors is a group or consortium [The CONTRAIS Study Group]. In addition to naming the author group, please list the individual authors and affiliations within this group in the acknowledgments section of your manuscript. Please also indicate clearly a lead author for this group along with a contact email address.

Author response:

The membership list, including affiliations for each member and the lead author with contact email address, is now available in the Supporting information according to the affiliations formatting guidelines.

Reviewers' comments:

Reviewer's Responses to Questions

Comments to the Author

1. Is the manuscript technically sound, and do the data support the conclusions?

Reviewer #1: Yes

Reviewer #2: Yes

2. Has the statistical analysis been performed appropriately and rigorously?

Reviewer #1: Yes

Reviewer #2: Yes

3. Have the authors made all data underlying the findings in their manuscript fully available?

Reviewer #1: Yes

Reviewer #2: Yes

4. Is the manuscript presented in an intelligible fashion and written in standard English?

Reviewer #1: No

Author response:

Additional copyediting has now been performed and is saved in revised manuscript with track changes.

Reviewer #2: Yes

5. Review Comments to the Author

Reviewer #1: As the statistical reviewer I will focus on methods and reporting. The analytical approaches are approprriate.

Major

1) It was not clear what the outcome is until the statistical analysis section. All the outcomes are listed with the primary outcome mentioned as not being reported in this work. I suggest have a section of outcomes that are evaluated in this work first, and then a second section of outcomes not used in this study.

Author response:

We have now changed according to the reviewers’ suggestions on page 7-9, line 160-209.

2) The power calculations are around the primary outcome, as is expected, but this is not an analysis of the primary outcome. Also the power calculations are around a massive effect (15% vs 45% failure in two of the groups) so anything smaller than that, this RCT will be underpowered to detect. So, overall, power is a major concern for this study, and findings need to be evaluated very concervatively. In particular, the interaction effect analyses will be grossly underpowered.

Author response:

We agree with the reviewer that exploratory analyses on secondary outcomes can be underpowered considering the clinical trials statistical power based on the primary analyses. Our primary power analysis was based on the NB-group success of 85% vs SSE-group or PA-group success of 55%. This is equivalent to an Odds ration difference between groups of 4.4 which is a large effect size. The aim of the current study is purely explorative without hypotheses of intervention effects on secondary outcomes. Our analyses primarily shows some small within- and between group changes over time. As stated on page 26 line 448-449, we have conducted a posteriori statistical power assessment on secondary outcomes based on recently reported effect sizes from other studies which did not exist when current trial was planned in 2012. The posteriori power assessment showed that we had adequate power for secondary outcomes. This decreases the likelihood of Type II errors and increases our confidence that our results would not significantly change in a larger cohort.

3) What do the univariable (not univariate) analyses add? I suggest dropping or moving to an appendix to simplify the paper.

Author response:

We have now changed the term univariate to univariable on page 11, line 247. Considering that this study aims to conduct an exploratory analysis, we find value in presenting and discussing both within-group and between-group analyses in the manuscript. The within-group analyses in this study help interpret our between-group analyses.

3) there is no information on missing data. why wasn't multiple imputation used as a main analytical approach? or were all data complete? please clarify in the methods section as per the CONSORT statement.

Author response:

On page 11, line 254-257 we have described the use of restricted maximum likelihood methods for intention-to-treat analyses. We have now added the word “imputation” to the sentence to clarify to the reader that imputation has been used. We have also added on page 11-12, line 257-259 “The data set contained missing data ranging from 15-23% per variable with the exception of the IPAQ sitting min/day variable with 44% missing data”. On page 12, line 262-265, a description of sensitivity analyses is presented which include a comparison of ITT and complete cases and reported in results on page 24, line 393-397. On page 26, line 452-453 a posteriori statistical power assessment is also mentioned as a methodology strength.

Reviewer #2: Thank you for writing the valuable article. Following I suggested some corrections on your article which hopefully make it better for your audiences.

Introduction section

Line 75: The verb “aims” came for “treatments”. If you want to say many kinds of treatment, you should use treatments with the related verb aim. Otherwise, you should use treatment with the verb aims.

Author response:

Thank you for the comment and we have now changed according to your suggestions in the introduction section on page 4, line 75 “Conservative treatments with a combination of braces and exercises for adolescents with idiopathic scoliosis (AIS) aim to prevent or limit curve progression”.

Conclusion section

Please describe better what results can derive from your study. You mentioned a general description of the study. It is recommended to describe more detail and don not use the abbreviation to better conclude the message of your study.

Author response:

We have now changed our conclusions according to your suggestions on page 4, line 66-68 and on page 27, line 489-491.

6. PLOS authors have the option to publish the peer review history of their article (what does this mean?). If published, this will include your full peer review and any attached files.

Do you want your identity to be public for this peer review? For information about this choice, including consent withdrawal, please see our Privacy Policy.

Reviewer #1: No

Reviewer #2: No

---

## [Decision Letter · Decision Letter 1]

21 Feb 2025

Trunk rotation, spinal deformity and appearance, health-related quality of life, and treatment adherence: Secondary outcomes in a randomized controlled trial on conservative treatment for adolescent idiopathic scoliosis

PONE-D-24-52488R1

Dear Dr. Dufvenberg,

We’re pleased to inform you that your manuscript has been judged scientifically suitable for publication and will be formally accepted for publication once it meets all outstanding technical requirements.

Kind regards,

Taher Babaee

Academic Editor

PLOS ONE

Additional Editor Comments (optional):

Reviewers' comments:

Reviewer's Responses to Questions

**Comments to the Author**

1. If the authors have adequately addressed your comments raised in a previous round of review and you feel that this manuscript is now acceptable for publication, you may indicate that here to bypass the “Comments to the Author” section, enter your conflict of interest statement in the “Confidential to Editor” section, and submit your "Accept" recommendation.

Reviewer #1: All comments have been addressed

Reviewer #2: All comments have been addressed

2. Is the manuscript technically sound, and do the data support the conclusions?

Reviewer #1: Yes

Reviewer #2: Yes

3. Has the statistical analysis been performed appropriately and rigorously?

Reviewer #1: Yes

Reviewer #2: Yes

4. Have the authors made all data underlying the findings in their manuscript fully available?

Reviewer #1: Yes

Reviewer #2: Yes

5. Is the manuscript presented in an intelligible fashion and written in standard English?

Reviewer #1: Yes

Reviewer #2: Yes

6. Review Comments to the Author

Reviewer #1: I am satisfied with the authors' responses and the resulting changes to the paper. note that post-hoc power calculations are relatively meaningless.

Reviewer #2: All comments have been addressed. It could be considered as an accepted article. I have no additional comments on this manuscript.

7. PLOS authors have the option to publish the peer review history of their article (what does this mean? ). If published, this will include your full peer review and any attached files.

**Do you want your identity to be public for this peer review?** For information about this choice, including consent withdrawal, please see our Privacy Policy .

Reviewer #1: No

Reviewer #2: No

---

## [Editor Report · Acceptance letter]

PONE-D-24-52488R1

PLOS ONE

Dear Dr. Dufvenberg,

I'm pleased to inform you that your manuscript has been deemed suitable for publication in PLOS ONE. Congratulations! Your manuscript is now being handed over to our production team.

Kind regards,

on behalf of

Dr. Taher Babaee

Academic Editor

PLOS ONE